# Thymoquinone Potentially Modulates the Expression of Key Onco- and Tumor Suppressor miRNAs in Prostate and Colon Cancer Cell Lines: Insights from PC3 and HCT-15 Cells

**DOI:** 10.3390/genes14091730

**Published:** 2023-08-30

**Authors:** Sofía Madeline Osorio-Pérez, Carolina Estrada-Meza, Luis M. Ruiz-Manriquez, María Goretti Arvizu-Espinosa, Aashish Srivastava, Ashutosh Sharma, Sujay Paul

**Affiliations:** 1School of Engineering and Sciences, Tecnologico de Monterrey, Campus Queretaro, Av. Epigmenio Gonzalez, No. 500 Fracc. San Pablo, Queretaro 76130, Mexico; 2School of Medicine and Health Science, Tecnologico de Monterrey, Monterrey 64700, Mexico; 3Department of Clinical Science, University of Bergen, 5021 Bergen, Norway

**Keywords:** microRNAs, thymoquinone, colorectal cancer, prostate cancer, HCT-15, PC3

## Abstract

Prostate cancer (PC) and colon cancer significantly contribute to global cancer-related morbidity and mortality. Thymoquinone (TQ), a naturally occurring phytochemical found in black cumin, has shown potential as an anticancer compound. This study aimed to investigate the effects of TQ on the expression profile of key tumor suppressor and onco-suppressor miRNAs in PC3 prostate cancer cells and HCT-15 colon cancer cells. Cell viability assays revealed that TQ inhibited the growth of both cell lines in a dose-dependent manner, with IC^50^ values of approximately 82.59 μM for HCT-15 and 55.83 μM for PC3 cells. Following TQ treatment at the IC^50^ concentrations, miRNA expression analysis demonstrated that TQ significantly downregulated miR-21-5p expression in HCT-15 cells and upregulated miR-34a-5p, miR-221-5p, miR-17-5p, and miR-21-5p expression in PC3 cells. However, no significant changes were observed in the expression levels of miR-34a-5p and miR-200a-5p in HCT-15 cells. The current findings suggest that TQ might exert its antiproliferative effects by modulating specific tumor suppressor and onco-suppressor miRNAs in prostate and colon cancer cells. Further investigations are warranted to elucidate the precise underlying mechanisms and to explore the therapeutic potential of TQ in cancer treatment. To the best of our knowledge, this is the first report regarding the effect of TQ on the miRNA expression profile in colon and prostate cancer cell lines.

## 1. Introduction

Prostate cancer (PC) and colon cancer are major types of cancer that significantly impact health and mortality globally [1]. Prostate cancer (PC) is a leading cause of morbidity and mortality in men, representing the second most commonly diagnosed malignancy worldwide and the fifth leading cause of cancer-related mortality [2,3]. In 2020, this equated to 1,414,249 newly diagnosed cases and 375,000 fatalities per year worldwide due to this disease [2]. It is anticipated that there will be 2,293,818 instances of this phenomenon up until the year 2040 [4]. Prostate cancer primarily impacts the male population within the age range of 45 to 60 years, and it is associated with various risk factors, such as familial predisposition, ethnicity, advanced age, obesity, and other environmental factors [5]. On the other hand, colon cancer, usually referred to as colorectal cancer (CRC), is a severe type of cancer and one of the most frequently found digestive malignancies worldwide [1]. This cancer ranks as the third most prevalent cancer in males and females, and it is the second leading cause of cancer-related deaths [6,7]. The number of CRC cases is expected to double globally in the following decades, reaching an estimated 2.5 million new cases globally in 2035 [8], with the most remarkable rise seen in less developed countries [9]. The cancer burden has risen in developed and developing nations over the years due to complex factors, including the aging and growth of populations, the acceleration of socioeconomic development, and shifts in the prevalence of risk factors [10]. Moreover, a growing occurrence of CRC among younger individuals is likely related to unhealthy dietary habits, inadequate physical activity, and elevated obesity rates [8,11].

Cancer is not only a very lethal and debilitating illness, but it also presents considerable medical challenges. Even though there has been advancement in the study of cancer, conventional treatment methods, including surgery, radiation therapy, chemotherapy, hormonal treatments, and targeted biological therapies, have a number of limitations, such as high prices, a lack of accessibility, and harmful side effects [12,13,14]. Thus, exploring novel approaches and alternatives to combat these diseases is crucial to enhance the treatment outcomes for individuals with cancer. There is an increasing focus on searching for and identifying potential secure and easily accessible anticancer compounds [15]. Epidemiological studies have consistently emphasized the chemopreventive properties of naturally occurring dietary compounds and their potential to naturally inhibit various diseases, including cancer [16]. Phytochemicals have been found to exhibit substantial anticancer attributes that target multiple stages of carcinogenesis, such as cell apoptosis and proliferation, cancer invasiveness, and disease metastasis [17].

Thymoquinone (TQ), identified as 2-methyl-5-isopropyl-1,4-benzoquinone, is a naturally occurring phytochemical present in the volatile oil derived from black cumin or black seed (*Nigella sativa*) [18]. It has been utilized in various traditional medicinal practices, predominantly in Arabian, Mediterranean, South Asian, and African regions [19]. TQ has been found to possess multiple pharmacological properties, such as antimicrobial, antioxidant, anti-inflammatory, antineoplastic, antidiabetic, antihypertensive, and neuro- and cardioprotective effects [18]. Thus, TQ exhibits the potential to regulate a wide range of physiological conditions, specifically diverse forms of cancer. It exhibits its anticancer effect by altering the expression and activation profile of crucial signaling molecules involved in the emergence of cancer hallmarks and their promoting traits. TQ significantly reduces the growth, progression, and metastasis of cancer by controlling the overlapping pathways responsible for cell proliferation, cell survival, cell death, invasive metastasis, and inflammatory signaling through a variety of mechanisms, including epigenetic regulation [20]. TQ has demonstrated its potential to trigger apoptosis, enhance DNA damage, repress the expression levels of DNA repair genes, and interfere with migration metastasis in both cultivated colon and prostate cancer cells and animal models [15,21,22,23]. In recent decades, the modulation of microRNA (miRNA) expression via TQ has been recognized as an innovative approach to addressing cancer [18]. 

MicroRNAs are a class of small, noncoding RNA molecules typically consisting of 20–24 nucleotides that play a crucial role in regulating gene expression at the post-transcriptional level, thereby contributing to the precise modulation of normal cellular physiology [24]. These molecules are known to significantly impact the pathogenesis of diverse cancers [1,25] through their regulation of genes involved in biological activities, such as cell differentiation, metabolism, cell growth, and cell death [26,27,28]. Given the significance of these processes in the tumorigenesis and progression of cancer malignancy, it is essential to understand the specific miRNAs involved [29]. In cancer, miRNAs have been found to play functional roles, acting as either tumor suppressors (TS) or tumor promoters (onco-miRs) [30]. TS miRNAs typically experience downregulation in cancer cells and play a crucial role in inhibiting cancer progression by suppressing the expression of oncogenes and genes that promote tumor growth [31]. For example, TS miR-34a has been identified in PC cells as a regulator of BCL2, resulting in the inhibition of its function and later leading to a reduction in the proliferation of tumor cells, an increase in apoptosis, and heightened sensitivity to chemotherapeutic agents [29]. Reports have stated that miR-34 exhibits down-regulation in cancerous prostate tissues compared to corresponding benign tissues and negatively impacts the essential metabolic process of prostate cancer [32,33]. In colon cancer, miR-34a also exhibits TS properties by regulating cell growth, migration, invasion, and apoptosis [33]. The progressive loss of miR-34a in colorectal cancer can have notable implications for the control of signaling pathways and gene expression networks. These changes ultimately contribute to the development and progression of tumors confirmed in diverse cell-based and animal study models [34,35,36]. Furthermore, overexpression of miR-200a in colon cancer, another TS miRNA, inhibits tumor growth and metastasis by modulating alternative signaling pathways [37].

On the contrary, the aberrantly elevated levels of oncogenic miRNAs in cancer contribute to the proliferation, motility, and metastasis of cancer cells through the suppression of numerous TS genes [38]. In human colon adenocarcinoma SW480 cells, the upregulation of miR-21, which serves as an oncomiR and is commonly observed to be overexpressed in colon cancer and other cancer types [39,40,41], has been shown to enhance cell proliferation and inhibit apoptosis significantly. The upregulation of oncomiR miR-21 has also been observed in PC3 cells and linked to various cellular processes, such as invasion, metastasis, cell proliferation, and apoptosis [42]. The microRNA miR-221 has been observed to be overexpressed in several types of epithelial cancers, such as breast, liver, bladder, pancreas, gastric, and colorectal cancer, melanoma, papillary thyroid carcinoma, and glioblastoma [43,44]. Its oncogenic properties are believed to be attributed to the suppression of the p27 tumor suppressor gene [43]. However, interestingly, both up- and downregulation of miR-221 has been reported in PC, indicating its potential tumor suppressor function, too [44]. The precise role of miR-221 in the process of tumorigenesis and the advancement of cancer remains a topic of debate. Likewise, the miR-17-92 cluster encompasses miR-17 and is associated with the modulation of numerous oncogenes and TSs that participate in diverse and contrasting pathways [45]. This phenomenon of duality effectively captures the intricate nature of cancer advancement, thereby facilitating the ability of miR-17-92 to precisely regulate the expression levels of genes that promote or inhibit apoptosis [45,46]. 

Due to the importance of the mentioned TS and onco-miRNAs, along with their mechanisms and reported frequency in various cancer types, we made the decision to investigate them in this study. However, a comprehensive understanding of how TQ impacts miRNA profiles in prostate and colon cancer is yet to be elucidated. This research aims to examine the antiproliferative properties of TQ and to assess its influence on the expression of significant TS and onco-miRNAs in both PC3 prostate cancer cells and HCT-15 colorectal adenocarcinoma cells. These cell lines were chosen because HCT-15 is a model system that is often used in cancer research to examine multiple aspects of colon cancer biology, such as tumor development, growth, toxicity, and drug resistance [47,48]. It has additionally been employed to examine the function and effectiveness of different natural extracts in combating colon cancer [49,50,51]. On the other hand, PC3 cells are considered a classic prostate cancer cell line widely used in drug research and development, making them a valuable model to investigate biochemical changes and to assess responses to chemotherapeutic agents in advanced prostate cancer cells [52,53].

## 2. Materials and Methods

### 2.1. Cell Lines 

The HCT-15 human colorectal adenocarcinoma and PC3 human prostatic adenocarcinoma cell lines were obtained from the American Type Culture Collection (ATCC). HCT-15 and PC3 cells were cultured in Dulbecco-modified Eagle medium (DMEM) with high glucose and low glucose, respectively, and supplemented with 10% fetal bovine serum (FBS). Cell cultures were incubated at 37 °C in a humidified 5% CO_2_ atmosphere. All reagents for cell culture were obtained from Sigma-Aldrich (Burlington, MA, USA).

### 2.2. Cell Viability Assay

The viability of cells was determined in the presence of thymoquinone (MedChemExpress, Monmouth Junction, NJ, USA). Thymoquinone (TQ) was dissolved in DMSO and resuspended in DMEM for dilutions. The final concentration of DMSO in the culture medium in both treatments and controls did not exceed 0.05%. HCT-15 and PC3 cells were seeded in 24-well plates at a density of 3.5 *×* 10^4^ (cells/well) and incubated for 48 h to obtain a confluency of 80%. The medium was replaced with 500 μL of fresh medium containing the specified TQ concentrations. TQ was administered at concentrations of 20, 40, 50, 70, 120, 330, and 420 μM for 24 h to HCT-15 cells. PC3 cells were treated with 0, 20, 25, 30, 45, 50, 60, 70, 80, 90, and 100 µM of TQ and incubated for 24 h. Cells not treated with TQ functioned as the control group. The viability was determined by 3-(4,5-dimethyl-2-thiazolyl)-2,5-diphenyl-2H-tetrazolium bromide (MTT) assay (Sigma-Aldrich, USA). Cells were treated with 300 µL of MTT solution (0.5 mg/mL in PBS) and were incubated for 4 h [54]. Subsequently, the medium was removed, and the formazan crystals were dissolved in each well with 600 µL of 80% ethanol to transfer in triplicate to a 96-well plate. Absorbance was measured at 570 nm in a microplate reader. A 100% viability was calculated using optical density values of the control cells. In cell viability studies, the IC50, or Half-Maximal Inhibitory Concentration, represents the concentration at which the compound decreases cell viability by 50%. A lower TQ IC50 implies that a smaller amount of the substance is needed to induce a decrease in cell viability. Each concentration was carried out in triplicate. We performed regression analyses to characterize the relationships between variables for each of the two cell lines separately. The choice of the regression model was based on the observed response patterns of each cell line. Linear regression was applied to cell line PC3 due to its apparent linear behavior, while exponential regression was chosen for cell line HCT-15, which exhibited exponential tendencies.

### 2.3. RNA Extraction and cDNA Synthesis

The HCT-15 and PC3 cell lines were seeded at a density of 3.5 *×* 10^4^ (cells/well) into 24-well plates. Once 80% confluency was reached, cells were treated using TQ (80 and 50 μM, respectively) for miRNA expression profiles and incubated for 24 h, 32 h, and 48 h, accordingly. Cells not treated with TQ functioned as the control group. The total RNA, including small RNAs, was then isolated and purified from cells using the miRNeasy Tissue/Cells Advanced Micro Kit (Qiagen, Hilden, Germany) according to the manufacturer’s protocol. RNA was quantified by NanoDrop One spectrophotometer (Thermo Scientific, Waltham, MA, USA). Subsequently, the MirX miRNA First-Strand Synthesis kit (Takara, Japan) was utilized to reverse transcribe the corresponding isolated RNA. The cDNA was diluted at a ratio of 1:10 in RNase-free water.

### 2.4. Computational Target Prediction of Studied Colon and Prostate Cancer miRNAs and Construction of Network Map

To predict the potential targets of the studied miRNAs, the miRTarBase (https://mirtarbase.cuhk.edu.cn, accessed on 5 January 2023) [55] was used to identify the most reported targets for miR-17-5p, miR-34a-5p, miR-21-5p, miR-200a-5p, and miR-221-5p in their respective type of cancer. miRTarBase was selected for predicting miRNA targets due to its decade-long reliability, encompassing over 13,000 articles, 27,000 target genes, and almost 19 million interactions. With natural language technology, it is a valuable resource for miRNA research in pathways, cancers, and therapies, aiding both researchers and drug development [56]. A network map was constructed based on the minimum folding free energy (MFE) values associated with the miRNA–target interaction. The network, consisting of miRNAs and their corresponding targets, was visualized using Cytoscape 3.2, a software tool (https://cytoscape.org/release_notes_3_2_0.html, accessed on 1 July 2023).

### 2.5. Experimental Validation and Expression Analysis of miRNAs and Target Genes by RT-qPCR

The Mir-X miRNA TB Green qRT-PCR kit (Takara, Japan) was used following the manufacturer’s protocol for PCR amplification performed in the Step One Real-Time PCR System (Applied Biosystems, USA). The primers for miR-34a, miR-21, miR-17, miR-200a, and miR-221 were designed using the miRbase (http://www.mirbase.org, accessed on 9 February 2023) and were supplied by T4 Oligo, Irapuato, Mexico. The miRTarBase (https://mirtarbase.cuhk.edu.cn, accessed on 5 January 2023) was used to identify the key miRNA targets. Primer design for amplifying the 3’UTR sequences of the expected miRNA target was conducted on the Benchling platform (https://benchling.com/, accessed on 8 May 2023). The specificity of primer binding was assessed using NCBI primer-BLAST. Quantitative PCR to determine the level of the mRNA targets was carried out by the TB Green Advantage qPCR premixes Kit (Takara, Japan) using U6 snRNA as a normalization control for miRNAs and β-actin for mRNAs/targets expression. The relative fold change data were calculated using the comparative Ct method, often known as 2^ΔΔCt^. Following the biological averaging strategy used in other research with real-time PCR, microarray, and RNA-Seq analysis, three pooled biological replicates and three technical replicates were used in qPCR experiments to upsurge statistical reliability [57,58,59]. 

### 2.6. Statistical Analysis

Paired Student’s *t*-tests were used to assess the statistical significance of the differences between the two groups obtained from the three biological replicates of each experiment. A *p*-value of less than 0.05 was used to determine statistical significance (* *p* < 0.05). The results were presented as the mean value ± the standard deviation (Mean ± SD). 

## 3. Results

### 3.1. Cell Viability of HCT-15 and PC3 Cells Following Thymoquinone Treatment and IC_50_ Value

This study included a cell viability assay using the MTT method to determine the impact of TQ on cancer cell proliferation across a range of concentrations (0–420 µM for HCT-15 and 0–100 µM for PC3) over a 24 h period. The control group consisted of cells that were left untreated with TQ. Figure 1 illustrates a significant reduction in cellular viability in both cell lines, where the IC_50_ value was estimated to be around 82.59 μM for HCT-15 (Figure 1A) and 55.83 μM for PC3 (Figure 1B). The growth of HCT-15 colorectal adenocarcinoma and PC3 prostatic adenocarcinoma cells was found to be inhibited in a dose-dependent manner by TQ. Moreover, even at greater doses, the TQ treatment of cells negatively impacted their growth.

### 3.2. miRNA Expression Analysis

HCT-15 colon cancer cells and PC3 prostate cancer cells were exposed to TQ’s experimental IC_50_ (80 µM and 50 µM, respectively) for 24 h. This research aimed to examine changes in the expression of miR-34-5p, miR-21-5p, and miR-200a-5p in HCT-15 and miR-34a-5p, miR-221-5p, miR-21, and miR-17-5p in PC3 cells subjected to TQ treatment compared to a control group. The results revealed that TQ significantly reduced the expression levels of miR21-5p in HCT-15 by 0.43-fold (*p* < 0.05) compared to the control group. However, no significant differences were observed in the expression levels of miR-34a-5p and miR-200a-5p between the TQ-treated and control groups, except a minor upregulation in miR-34a-5p (1.021-fold) and miR-200a-5p (1.2-fold) expression levels (*p* > 0.05) for HCT-15 cells (Figure 2A). Based on these findings, miR21-5p, which showed statistically significant deregulation and an inhibitory effect, was further investigated to assess its expression levels at 48 h. The results demonstrated that TQ continued to significantly decrease miR21-5p expression in HCT-15 cells by 0.36-fold (*p* < 0.05) compared to the control group. Additionally, there was a significant decrease of 0.11-fold (*p* < 0.05) in miR21-5p expression from 24 to 48 h, indicating a time-dependent inhibitory response of miR21-5p (Figure 3A). The findings for PC3 cells indicate that TQ significantly impacted the upregulation of miR-34a-5p expression levels, with a 4.45-fold increase observed compared to the control group (*p* < 0.05). Furthermore, TQ exhibited a significant increase in the PC3 expression levels (*p* < 0.05) of miR-221-5p (2.57-fold), miR-17-5p (11.85-fold), and miR-21-5p (10.38-fold), as illustrated in Figure 2B. Following the findings of the PC3 cell line, a further study was conducted to assess the expression levels of miR-34a-5p after a 32 h time frame. The outcome of this study indicated that TQ administration resulted in a reduction of expression after 32 h, with a decrease of 0.31-fold compared to the control group (Figure 3B). 

### 3.3. Computational Target Prediction of Studied Colon and Prostate Cancer miRNAs

A comprehensive analysis revealed a total of 32 significant predicted targets associated with the investigated miRNAs in the context of human prostatic adenocarcinoma and human colorectal adenocarcinoma (Figure 4). Notably, in the network map, BCL2 and PTEN exhibited a higher number of connections, as they were targeted by four of the five miRNAs investigated. Consequently, it was deemed appropriate to proceed with the analysis focusing on these two particular targets. This approach allowed us to observe the distinction between an oncogenic gene (BCL2) and a tumor suppressor gene (PTEN).

### 3.4. Target Genes Expression Analysis

To validate the expression of target genes BCL2 and PTEN in colon and prostate cancer, qPCR analysis was performed for both TQ-IC_50_-treated and untreated cells at 24 h, similar to the miRNA validation process. For HCT-15, a noticeable change in the expression levels of these genes was observed when comparing the treated and untreated cells (refer to Figure 5). Notably, the downregulation of the BCL2 (*p* < 0.05) and PTEN (*p* > 0.05) genes was observed, showing a possible correlation with the expression of the corresponding miRNAs. In the case of PC3 cells, the expression levels of the targets were upregulated with a high standard deviation, giving inconclusive results compared with their miRNA expression. Due to the inconsistency in PC3 cells, the results are not presented, and there is a need to evaluate other implicated targets. 

## 4. Discussion

Currently, extensive research aims to mitigate the incidence of cancer-related mortality and morbidity. Furthermore, exploring alternative strategies to combat cancer effectively is crucial [5,9]. In recent times, there has been a growing interest among researchers in plant-based medicines and their constituent compounds. The focus on utilizing plant extracts, such as TQ derived from *Nigella sativa*, as a potential source of anticancer compounds offers promising prospects [23]. Several research studies have demonstrated that TQ possesses the ability to suppress cell proliferation and enhance apoptosis in malignant cells [60,61,62,63]. Prior research has demonstrated that TQ can induce dose-dependent cell death in colon and prostate cancer cell models, including the HCT-116 human colorectal carcinoma cell line, the HT-29 human colorectal adenocarcinoma cell line, and the PC3 prostatic adenocarcinoma cell line [60,64,65].

In order to investigate the cytotoxic impact of TQ on cancer cells, a viability assay was conducted on both cell lines (HCT-15 and PC3). The findings indicate that TQ treatment significantly reduced cell viability in a dose- and time-dependent manner. Furthermore, the IC_50_ values obtained for TQ treatment over a period of 24 h were 82.54 µM for HCT-15 and 55.83 μM for PC3 cell lines. Notably, a significant reduction in cell viability was observed in the concentration range of 50 to 210 μM for HCT-15, which is consistent with the findings in the HCT-116 cell line reported by Özkoç [64], who observed a marked decrease in cell viability in the range of 60 to 200 µM, and the IC_50_ value for TQ was determined to be 68 µM at 24 h [64]. Additionally, Saffari-Chaleshtori [60] showed a notable decrease in cell viability in a range of 20 to 70 μM for PC3 cell lines, reporting an IC_50_ of 40 μM, which is consistent with our assay obtaining an IC_50_ of 55.83 μM. This similarity in the concentration range and the IC_50_ values supports the notion that TQ exerts a comparable inhibitory effect on the proliferation of colon and prostate cancer cell lines. 

The correlation between miRNA regulation and TQ’s ability to prevent the advancement of tumors has been observed [18]. Hence, the capacity to regulate the expression of particular miRNAs could lead to remarkable implications concerning the treatment and management of colon and prostate cancer [15,22]. It has been established that TQ has the ability to block tumor growth by modulating miRNAs, which regulate signaling pathways implicated in the pathogenesis of cancer cells, such as growth, metastasis, angiogenesis, cell death, and epigenetic mechanism. The anticancer impact of TQ is mediated via miRNA control of a number of signaling pathways, including P53, PCNA, cyclin D1, BCL2, NF-B, TWIST, ZEB, eEF-2K, PI3K/Akt, and Src/Fak [18,23]. TQ has been documented as controlling miRNA activity in various cancer types. For instance, in pancreatic cancer, it enhances miR-23-1 and miR-101 levels, while in breast cancer, it boosts miR-361 and miR-34a expression. Conversely, it reduces miR-206 but increases miR-16. In liver cancer, it raises miR-16 and miR-375 levels, and in leukemia, it elevates miR-29b expression [18]. Nevertheless, the impact of TQ on miRNA patterns in cases of colon and prostate cancer is elusive.

Currently, miR-34a is the most extensively researched TS miRNA, exhibiting the ability to induce apoptosis and cell cycle arrest [66]. The downregulation of miR-34a has been implicated in human colon and prostate cancer, leading to increased cell proliferation, migration, and invasion, while its overexpression facilitates cell apoptosis in in vitro studies [18,65,67]. Numerous cellular and animal-based investigations support the tumor-inhibiting role of miR-34a and suggest that reinstating miR-34a expression could serve as a viable therapeutic approach [68]. In line with these findings, the present results demonstrate a slight increased miR-34a-5p expression in colon cancer cells HCT-15 and a significant increase in prostatic adenocarcinoma cells PC3 following exposure to TQ. The results indicate that TQ could have a significant effect on the inhibition of cell proliferation and the expression level of miR-34a-5p in PC3 cells, exhibiting a 4.45-fold rise compared to the control group (*p* < 0.05) within 24 h. However, after 32 h, the expression starts to decrease. Notably, the downregulation at 32 h is not significant, implying a potential return to the basal state. This possibly suggests a temporary effect; however, further research is necessary to ascertain the dynamics of this response. According to Thermo Fisher, the doubling time of PC3 cells is approximately 33 h. To ensure that the doubling time did not influence miRNA expression, we opted to evaluate miRNA expression levels prior to the doubling time. This choice aimed to minimize the impact of cell division on miRNA expression, allowing us to focus on the effects of treatment rather than cell proliferation dynamics. By selecting time points before the doubling time, we aimed to isolate the treatment-related changes in miRNA expression from the influence of cell division rates [69]. However, additional research is necessary to assess whether TQ demonstrates a greater significance in terms of TQ concentration as opposed to exposure duration, as indicated in prior investigations [70]. Meanwhile, mild overexpression of 34a-5p in HCT-15 demands further investigation to fully understand how it contributes to colon carcinoma development. Given the presence of various feedback regulatory mechanisms of miR-34a, it is imperative to consider the effects of these mechanisms on both the downstream targets and upstream regulators of miR-34a in the context of therapeutic introduction or re-expression of miR-34a [68]. In this approach, a negative correlation was observed between miR-34a and its target BCL2 in HCT-15 cells based on the functional role of miR-34a as a potent tumor suppressor, as it selectively targets mRNAs involved in antiapoptotic mechanisms, exemplified by its regulation of BCL2 expression [71,72]. However, due to the absence of significant regulation of mir-34a after TQ treatment, this correlation is not conclusive in the present analysis. Thus, as previously stated, it is imperative to conduct further research on the regulation of this miRNA before advancing to another analysis. Regarding PC3 cells, it would be intriguing to assess the expression levels of potential targets of miR-34a that exhibit notable expression, such as BCL2 and other relevant candidates. miR-34a has demonstrated the capability to impede proliferation, migration, motility, and invasion, while promoting apoptosis and senescence through its interactions with some key molecular targets, including SATB2, MYC, E2F3, NOTCH1, MET, AXL, and CDK4/6 [73,74,75,76,77].

A similar pattern was observed with the TS miR-200a-5p, where TQ treatment induced its upregulation in HCT-15 cells. Although this study did not find a significant effect, previous research has shown that overexpression of miR-200a-3p mediated by transfection with a miR-200a-3p mimic functionally suppresses the proliferation, migration, and invasion of colon cancer cells in DLD1 and SW480 cell lines [78]. Additionally, compounds like ursolic acid have considerably raised the expression levels of miR-200a in HCT116 and HCT-8 cells, emphasizing the role that miR-200a plays in developing colon cancer [79]. The functional impact of miR-200 family members is mediated through their targeting of multiple mRNAs involved in the proliferation of cancer cells. It has been reported that the miR-200 family can selectively target PTEN, a crucial suppressor of the PI3K/AKT pathway, which facilitates proliferation, migration, invasion, and epithelial–mesenchymal transition [80]. However, as mentioned for mir-34a, due to the absence of significant regulation of this miRNA after TQ treatment, this correlation is inconclusive, and further investigation into the regulation of these miRNAs is essential before proceeding with further analyses. 

The upregulation of miR-221 has been observed in several types of epithelial tumors, such as breast, liver, bladder, pancreas, gastric, and colorectal cancer, melanoma, papillary thyroid carcinoma, and glioblastoma, appearing to be an oncogenic miRNA [43]. Both upregulation and downregulation of miR-221 have been reported in PC, as documented in several studies [43,44]. Previous studies have indicated that the upregulation of miR-221-5p results in decreased proliferation of prostate cancer cells by suppressing cell cycle regulatory proteins, implying that miR-221-5p may exhibit TS properties [43]. Supporting these findings, this study demonstrated that overexpression of miR-221 led to decreased cell proliferation after TQ treatment. A significant increase in the expression levels (*p* < 0.05) of miR-221-5p was observed, with a 2.57-fold increase compared to the control group. Research suggests that miR-221-5p has the ability to target various pathways and to function as either a TS or oncogenic miRNA, depending upon the specific cellular and experimental circumstances [43]. Some of the potential targets that this miR-221 is targeting and that might attract attention for validation due to the significant result of this miRNA in prostate cancer PC3 cells include targets that play important roles in reducing angiogenesis, inhibiting cell viability, regulating the cell cycle, restraining invasion and proliferation, and inducing apoptosis both in vitro and in vivo. Noteworthy candidates include KIT, MDM2, TIMP/3, CDKN1C, and CDKN1B [81,82,83,84]. Thus, comprehending the role of miR-221 in the context of PC may facilitate the discovery of novel interactions among signaling pathways that enable the progression of PC. 

MiR-17-5p exhibited TS properties in PC cells, despite being reported as an oncogenic miRNA in other cases. TQ-treated PC3 cells were significantly upregulated by an 11.85-fold increase compared to the control group (*p* < 0.05). Antiproliferative effects were observed upon miR-17-5p overexpression. Previous studies have reported decreased levels of miR-17-5p and miR-17-3p in both prostate cancer cell lines and tissues, providing further evidence for the aforementioned findings [45,46]. Comprehensive investigations are necessary to fully understand the mechanistic functions of the miR-17-5p in the regulation of specific pathways that result in the suppression of cellular proliferation.

Accumulating evidence indicates that miR-21 is an extensively investigated upregulated oncogene in various cancer types and that it is associated with numerous biological processes of cancer [85,86,87]. The data collected in other studies have provided evidence that miR-21 is prominently upregulated in colon cancer cells, such as SW480, HT29, and HCT116. Moreover, eliminating this specific miRNA has been correlated with decreased proliferation and enhanced apoptosis in these cells [85,88,89]. Phytochemicals, such as celastrol and curcumol, have also been documented to downregulate miR-21, reducing colon cancer cell viability in in vitro studies [39,90], supporting the notion that natural phytochemicals possess the ability to modulate miR-21 expression and impact cancer cell behavior [24]. Consistent with this, the administration of TQ to HCT-15 colon cancer cells resulted in the inhibition of cell proliferation and a notable reduction in the expression of miR-21-5p compared to the control group of untreated cells. Furthermore, it was sought to analyze the effect that TQ had on the profile of this miRNA over time. TQ exhibited a significant decrease in miR-21-5p levels between the 24 h and 48 h time points, indicating a time-dependent effect. The HCT15 cell line was evaluated at 24 and 48 h due to the frequency with which this cell line is investigated at these periods, as well as the results that have been achieved with TQ; for example, the finding exhibited in this research aligns with similar observations reported in other articles using colon cancer cells, such as HCT-116, where they employed diverse treatment durations of 24 and 48 h and demonstrated a significant increase in apoptosis-mediated cell death over time [91,92]. In addition, it has been observed that treatment with TQ at its IC_50_ concentration resulted in a noticeable decrease in viable cancer cells after 24 and 48 h [93]. miR-21 focuses on crucial tumor suppressor genes and genes linked to cancer development, like PTEN. Consequently, when miR-21 is downregulated, it has the potential to relieve PTEN suppression, potentially reinstating its tumor suppressor activities [94,95]. Nevertheless, it is important to note that the results of this investigation, by maintaining suppressed PTEN expression, do not necessarily suggest an impact of PTEN-mediated TQ on cell viability. Conversely, miR-21 binding can stimulate the expression of BCL2, resulting in decreased apoptosis and enhanced proliferation [94]. Consequently, a decrease in miR-21 levels leads to a decrease in BCL2 expression [95], as evidenced by the findings of this study. Furthermore, other potential miR-21 targets that could be validated include those linked to cellular processes, like cell proliferation, migration, and invasion processes, such as RECK, PDCD4, TPM1, MSH2, BTG2, and SPRY2 [94,96,97,98,99].

In prostatic adenocarcinoma cells, miR-21-5p expression showed a significant upregulation of 10.38-fold compared to the control (*p* < 0.05). In addition, research using prostate models indicates that miR-21 is not a common or significant factor in PC, implying that its sole targeting does not constitute a useful therapeutic strategy in such a condition [100]. Elucidating the underlying molecular mechanisms by which TQ and other compounds modulate miRNA expression would provide valuable insights into their therapeutic cancer potential. Understanding the downstream mRNA targets, protein levels, and broader regulatory networks affected by miRNA modulation can aid in developing targeted interventions that disrupt the pro-cancer effects. 

Although this research could demonstrate the potential of thymoquinone (TQ) as an anticancer compound in prostate cancer (PC) and colon cancer, certain limitations warrant consideration. This study’s focus on a specific set of cell lines may limit the broader applicability of the findings. Furthermore, the observed miRNA expression changes, while intriguing, might require reassessment and mechanistic explanations, warranting further investigation to strengthen the conclusions. The inability of this study to induce significant changes in miRNA and its targets in certain cell lines underscores the complexity of the effects of TQ. Furthermore, although the in vitro results are promising, the absence of clinical and in vivo data makes direct translational implications difficult. Recognizing and addressing these limitations improves the balance of the manuscript and sets a clear path for future research in harnessing the therapeutic potential of TQ.

## 5. Conclusions

The findings of our study indicate that the phytochemical TQ exhibits antiproliferative properties by inhibiting the proliferation of HCT-15 and PC3 cancer cells. Additionally, TQ alters the expression of several key TS and onco-microRNAs, including miR-200a-5p, miR-221-5p, miR-17-5p, miR21-5p, and miR-34a-5p. The significant influence of miR-34a-5p, miR-221-5p, miR-17-5p, and miR21-5p suggests its potential as a therapeutic target for managing prostate and colon cancer, respectively. These findings promise to advance therapeutic interventions using natural remedies to modulate miRNA expression and to regulate other crucial miRNAs involved in colon and prostate cancer pathogenesis. Further experiments are needed to fully understand the anticarcinogenic role of TQ in colon and prostate cancer through the modulation of miRNAs. 

## Figures and Tables

**Figure 1 genes-14-01730-f001:**
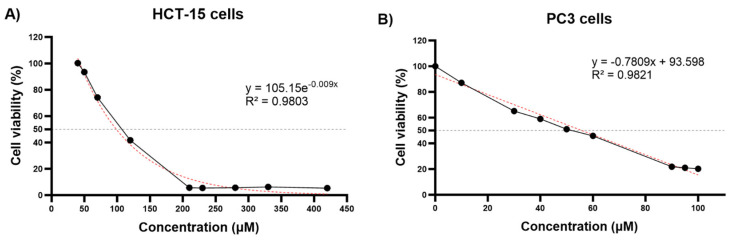
TQ effect on PC3 and HCT-15 cell viability. (**A**) The impact of concentrations ranging from 20 to 100 μM of TQ on the viability of PC3 cells for 24 h was examined. (**B**) Concentrations ranging from 20 to 420 μM of TQ were also evaluated for their impact on the viability of HCT-15 cells for 24 h. The viability of both cell lines was assessed using the MTT assay. The results were compared to a control group and were obtained from at least three replicates, with the values presented as the mean ± standard deviation.

**Figure 2 genes-14-01730-f002:**
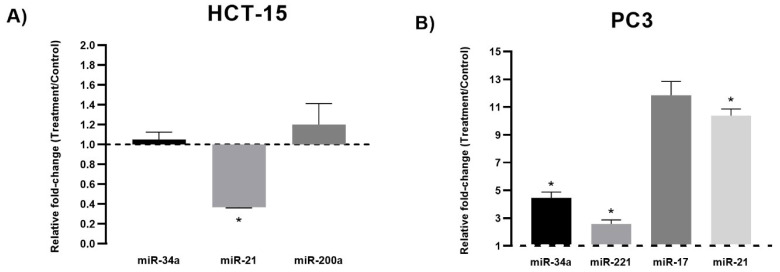
MiRNA expression profile of HCT-15 and PC3 cells treated with TQ. Real-time quantitative PCR analysis was conducted on (**A**) HCT-15 and (**B**) PC3 cells exposed to TQ at concentrations based on their IC50 value (80 µM and 50 µM, respectively) for 24 h. The U6 gene was used as a control for normalization. Each bar graph represents the mean values of relative fold change ± standard deviation obtained from triplicate assays. Asterisks indicate significant differences (*p* < 0.05).

**Figure 3 genes-14-01730-f003:**
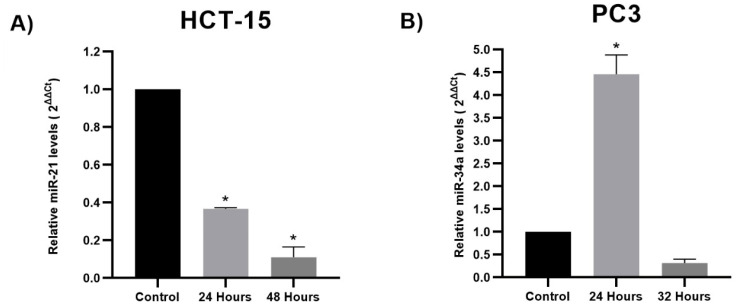
Time-dependent downregulation of miR-21-5p and miR-34a-5p expression in HCT-15 and PC3, respectively. (**A**) Downregulated miR21-5p expression was significantly detected in colon cancer cells HCT-15 at 24 h and 48 h after 80 μM TQ exposure. (**B**) Significant upregulation of miR-34a-5p at 24 h and subsequent downregulation after 32 h at a 50 μM TQ administration. Each bar graph represents the mean values of relative fold change ± standard deviation obtained from triplicate assays. Asterisks indicate significant differences (*p* < 0.05).

**Figure 4 genes-14-01730-f004:**
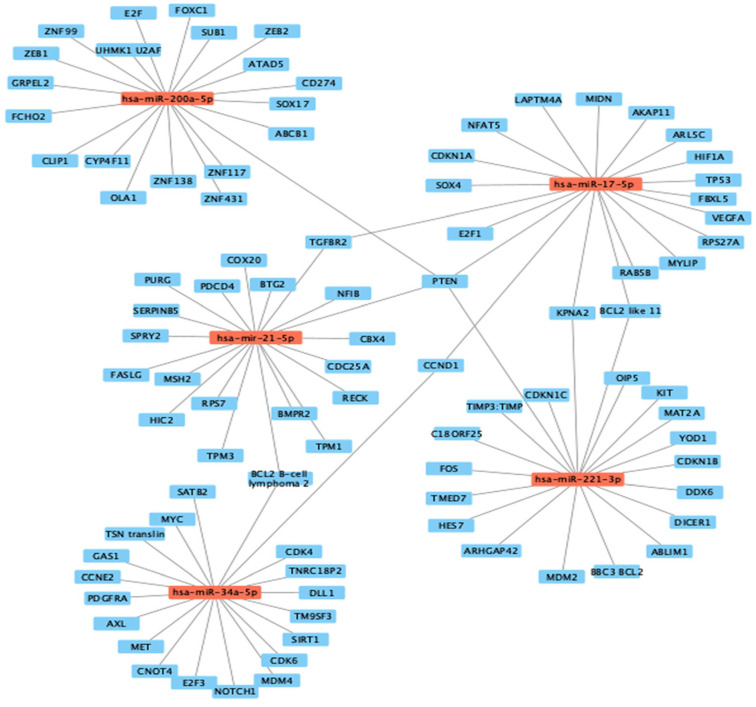
MFE-based network interaction map of potential targets of the studied miRNAs in colon and prostate cancer.

**Figure 5 genes-14-01730-f005:**
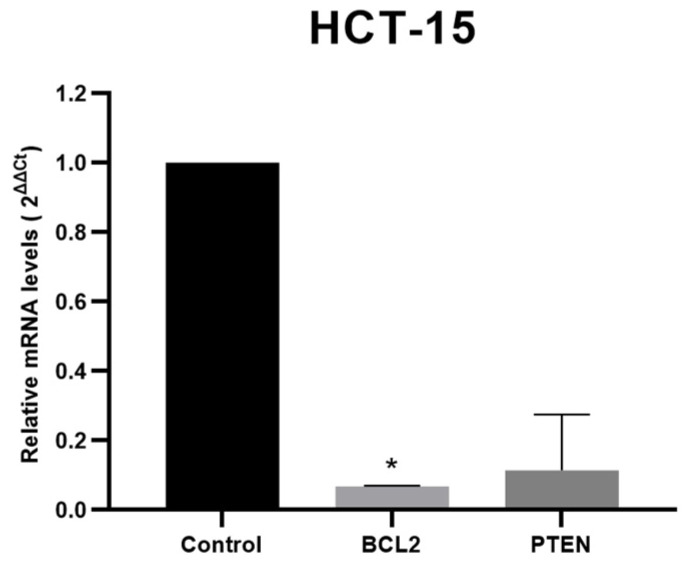
Validation of target genes by qPCR. The relative expression values were obtained to indicate the relative alteration in the mRNA expression levels of BCL2 and PTEN in colon cancer cells HCT-15 at 24 h after 80 μM TQ exposure. These values were normalized to β-actin. Each bar graph represents the mean values of relative fold change ± standard deviation obtained from triplicate assays. Significant differences (*p* < 0.05) are indicated by asterisks.

## Data Availability

The data that support the findings of this study are available on request from the corresponding author.

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
