# Peer review of "Thymoquinone Potentially Modulates the Expression of Key Onco- and Tumor Suppressor miRNAs in Prostate and Colon Cancer Cell Lines: Insights from PC3 and HCT-15 Cells"

_genes, 2023, doi:10.3390/genes14091730_

Round 1

Reviewer 1 Report

Osorio-Perez et al. investigated Thymoquinone (TQ) from black cumin as an anticancer agent for prostate and colon cancer. Both types of cancer have significant global morbidity and mortality rates. The researchers examined the effects of TQ on key onco- and tumor suppressor microRNAs (miRNAs) in PC3 prostate cancer cells and HCT-15 colon cancer cells. The results indicate that TQ inhibited the growth of both cell lines in a dose-dependent manner, with specific IC50 values for each cell line. Furthermore, after treatment with TQ at IC50 concentrations, they observed significant downregulation of miR-21-5p in HCT-15 cells and upregulation of miR-34a-5p, miR-221-5p, miR-17-5p, and miR-21-5p in PC3 cells. TQ showed inhibitory effects on both cancer cell lines in a dose-dependent manner, regulating specific miRNAs. This pioneering study highlights TQ's potential in cancer treatment, offering promising insights for further research. The research paper is professionally written and fits the scope of the journal. Impressive work!

1.     Introduction is well written overall with good introduction to both cancer types. However, in the second paragraph (Line 52 onwards) it would be good to include more information about how the study developed to lead into the next paragraph and discussing more about current treatments.  

2.     Some information about the mechanistic background for TQ will be helpful in the introduction.

3.     Good introduction to miRNAs written by the authors!

4.     Can the authors provide more details of why only the mentioned miRNAs were studied?  Can the authors include some bioinfo data from TCGA and CCLE databases for the studied cancer types to show the levels of these miRNAs/validated genes. This information is critical to understand the broader view of cancer types and not be restricted to cell lines as cell lines do not fairly represent the entire cancer subtype.

5.     In the methods section, line 136, it mentions 48 hours. However, in the results, 24 hours is mentioned. Could you please check and rectify?

6.     Why did the authors select only HCT-15 and PC3 lines? Can these results be validated in at least one more cell line for each cancer type?

7.     Figure 1 can be changed to represent survival at 100% and subsequent levels for different doses. This will significantly improvise the representation for easy evaluation of results. Also, mark IC50 value in the graph for both cancer types.

8.     How do the authors explain the different trend of miRNA-21 in both cancer types upon TQ treatment.

9.     Figure 3 shows different time points for each cancer type. Why were these time points selected.

10.  Is it possible to efficiently perform analysis on other targets in the study that have high number of connections.

11.  For target validation, can the authors look for other targets from the interaction map.

12.  More limitations need to be addressed.

13.  Starting section of the discussion overlaps with introduction. Also, elaborate on why there is a need for alternate strategies.

14.  Paragraph starting from line 330: It would be nice to include mechanistic background of miRNA found up or downregulated by TQ treatment.

Author Response

Comment 1.     Introduction is well written overall with good introduction to both cancer types. However, in the second paragraph (Line 52 onwards) it would be good to include more information about how the study developed to lead into the next paragraph and discussing more about current treatments. 

Response 1. We are very thankful to the Hon´ble reviewer for his/her useful suggestions and recommendations. We have now added more information in the revised MS about conventional treatments and the importance of alternative treatments (lines 52 to 56).

Comment 2.     Some information about the mechanistic background for TQ will be helpful in the introduction.

Response 2. We have now added information in the revised MS regarding the mechanistic background for TQ (lines 72-78).

Comment 3.     Good introduction to miRNAs written by the authors!

Response 3. Many thanks.

Comment 4. Can the authors provide more details of why only the mentioned miRNAs were studied?  Can the authors include some bioinfo data from TCGA and CCLE databases for the studied cancer types to show the levels of these miRNAs/validated genes. This information is critical to understand the broader view of cancer types and not be restricted to cell lines as cell lines do not fairly represent the entire cancer subtype.

Response 4. Due to the preliminary nature of our study (that´s why we have submitted this article as a brief report), we focused on a selected group of miRNAs based on extensive literature research. These miRNAs were chosen for their frequent association with the studied cancer types and potential interactions with thymoquinone. We acknowledge the importance of including more diverse data sources, but our findings contribute a unique perspective as the first evaluation of thymoquinone's effects on miRNA profiles in these cancer types. Nevertheless, we will keep in mind the valuable suggestion of the esteemed reviewer and will include them in our future research.

Comment 5.     In the methods section, line 136, it mentions 48 hours. However, in the results, 24 hours is mentioned. Could you please check and rectify?

Response 5.  We greatly appreciate the reviewer's observation. It has already been mentioned in the cell viability section of the original MS (line 155 of the original MS) that cells were initially incubated for 48 hours to obtain 80% confluency. After reaching the confluency, cells received the TQ treatment at different times for each cell line, for which a brief clarification was added at line 177 of the revised MS.

Comment 6.     Why did the authors select only HCT-15 and PC3 lines? Can these results be validated in at least one more cell line for each cancer type?

Response 6. These cell lines were used because they are model cell lines for evaluating multiple biological mechanisms in their respective cancers. We´ve now added a small justification and explanation regarding the rationale behind selecting these cell lines (lines 133-140) of the revised MS.

Since the current journal is providing the platform to publish promising preliminary data as a ´brief report´ and since we found our preliminary data quite promising, we decided to submit it as a brief report to keep our priority. However, we will continue with this study for a deeper understanding and will keep in mind the valuable suggestion of the esteemed reviewer. We hope the Hon´ble reviewer will consider this.

Comment 7.     Figure 1 can be changed to represent survival at 100% and subsequent levels for different doses. This will significantly improvise the representation for easy evaluation of results. Also, mark IC50 value in the graph for both cancer types.

Response 7. We've now incorporated the IC50 values into the respective graphs for clarity. The calculated IC50 for each cell line is already presented in lines 228 and 229 of the original MS.

Comment 8.     How do the authors explain the different trend of miRNA-21 in both cancer types upon TQ treatment.

Response 8.  The divergent behavior of miRNA-21 in different cancer types following TQ treatment can be attributed to the complex nature of microRNA regulation within distinct cellular contexts. MicroRNAs like miRNA-21 can exhibit versatile roles, functioning as either oncogenes or tumor suppressors based on their target genes and the specific signaling pathways they modulate. Additionally, the specific downstream effects of miRNA-21 modulation could differ between cancer types due to the intricate network of interconnected pathways involved in cell proliferation, apoptosis, migration, and other cellular processes. Even slight variations in these pathways can lead to distinct outcomes in response to TQ treatment. Further research can clarify these differences and shed light on the underlying mechanisms.

Comment 9.     Figure 3 shows different time points for each cancer type. Why were these time points selected.

Response 9. A justification of using different time points for each cancer type is now added in the revised MS, HCT-15 (lines 453-456) and PC3 (lines 378-384).

Comment 10.  Is it possible to efficiently perform analysis on other targets in the study that have high number of connections.

Response 10. As we already mentioned, due to the preliminary nature of the results, the current stage of the study might not provide the necessary data and insights required to efficiently delve into analyses of highly connected targets.

Comment 11.  For target validation, can the authors look for other targets from the interaction map.

Response 11. Certainly, exploring additional targets from the interaction map for target validation is a promising avenue. However, before proceeding, it's essential to reevaluate the interaction between thymoquinone and the miRNAs. Understanding the intricacies of this interaction will provide valuable insights into the underlying mechanisms and guide the selection of potential targets for further analysis. This strategic approach ensures that the subsequent target validation is grounded in a solid understanding of the interactions, leading to more accurate and meaningful results.

Comment 12.  More limitations need to be addressed.

Response 12. More limitations of this work are now addressed (lines 479 to 489) in the revised MS.

Comment 13.  Starting section of the discussion overlaps with introduction. Also, elaborate on why there is a need for alternate strategies.

Response 13. The need for alternative strategies is now also addressed in the revised MS (lines 52-56), describing the limitations of conventional treatments that generate a need for other options.

Comment 14.  Paragraph starting from line 330: It would be nice to include mechanistic background of miRNA found up or downregulated by TQ treatment.

Response 14. The mechanism of thymoquinone in miRNAs from different cancers and its regulation are now added in the revised MS (lines 352-363).

Author Response

Comments for authors: This study adds to our existing knowledge of anti-proliferative potential of TQ and also provide evidence of miRNA and their target gene modulation after TQ treatment. However, there are many caveats in this study that should be addressed before publishing this report,

Major:

Comment 1. The study keeps shifting between the two different cancer cell lines but lacks strength and consistency of results in any of the two cell lines. The authors have shown overexpression of miR-34a-5p and other miRNAs in PC-3 cells (Fig 2B) and downregulation of their targets in HCT-15 cells (Fig 5), which does not demonstrate a negative correlation of miRNAs and target genes because it is not from the same cell lines. To my understanding, the levels of miR-34a-5p are not observed upregulated (only ~1.01 folds) and also miR-200a are very slightly upregulated (not significant) in HCT-15 cells treated with TQ (Fig 2A). Therefore, any further study on target of these miRNAs in HCT-15 cells should only be performed after re-evaluating the levels of these miRNAs in TQ treated HCT-15 cells.

Response 1. We are very thankful to the Hon´ble reviewer for his/her insightful feedback. It's important to note that the nature of the results in this study is preliminary (Since the current journal is providing the platform to publish promising preliminary data as a ´brief report´ and since we found our preliminary data quite promising, we decided to submit it as a brief report to keep our priority. However, we will continue with this study for a deeper understanding and will keep in mind the valuable suggestion of the esteemed reviewer. We hope the Hon´ble reviewer will consider this.) and the observed expression patterns of miRNAs in PC-3 cells and their targets in HCT-15 cells were explored independently. The article does not establish a direct correlation between these two sets of results due to the distinct cancer cell lines involved. Additionally, the inconsistency of target results in PC-3 cells, as noted in lines 313-316, led to their exclusion from the original MS. We appreciate the input of the learned reviewer, which helps us ensure the proper contextualization of our study's findings. In the case of the regulations for HCT-15 cells of miR-34a-5p and miR-200a-5p, it was already specified in lines 248-252 of the original MS that these were not significant regulations. Therefore, the following time-dependent analysis was only performed for miR-21-5p. The need to reassess the regulation of miR-34a and miR-200a and the inconclusive relationship with their targets is now mentioned (lines 395-399 and 413-415, respectively) in the revised MS.

Comment 2.        The two common target genes, BCL2 and PTEN (Fig 5) have tissue specific roles in cancer progression. While BCL2 is a known anti-apoptotic gene (oncogene), PTEN is frequently observed as a tumor suppressor gene in both colorectal and prostate cancers [1,2]. However, the authors have shown reduced levels of PTEN after TQ treatment. In light of previous literature, these results do not imply a PTEN mediated effect of TQ on cell viability. In case the authors find a strong piece of literature that can support their hypothesis, they can cite the information.

Response 2. In the revised MS, we have now incorporated a clarification (lines 460-466) to elucidate the role of PTEN. Furthermore, we have highlighted that TQ does not necessarily impact its expression or mechanism.

Comment 3.  It is unclear why the authors have used different timepoints for studying the expression of miRNAs in HCT-15 and PC-3 cells. They have evaluated 0, 24 and 48h results in HCT-15 and 0, 24 and 32 h results in PC-3 cells that is inconsistent. Also, the levels of miR-34a-5p after 24 h and 32 h treatment are observed as very drastic i.e. from 4.45 folds to 0.31 folds and therefore it cannot be conclusively stated if TQ treatment upregulates or downregulates miR-34a-5p in PC-3 cells.

Response 3. Using different times in the treatment, we seek to evaluate a time-dependent response regulating miRNAs for this compound. A justification for the same is now added in the revised MS, HCT-15 (lines 453-456) and PC3 (lines 378-384). We appreciate the esteemed reviewer for his/her attention to detail. A clarification has now been added to the revised manuscript to address the observed drastic changes in miR-34a-5p levels after 24 and 32 hours of treatment (lines 376-378).

Minor:

Comment 1.  The figures are blur and quality of all the figures need to enhanced for better legibility.

Response 1. We tried our best to improve the quality of the figures in the revised MS. We request the esteemed reviewer to check the figures in the Word file of the revised MS.

Comment 2.  The definition of IC50 as mentioned in methods (lines 147-149) is misleading and should be corrected to “Half maximal inhibitory concentration” as followed in the experiments. In addition to the IC50 value, it can be more informative if authors can point out (or provide the values of) TQ values for maximum inhibition on respective cell lines.

Response 2. The definition of the term "IC50" is now corrected in the revised manuscript, accurately referring to it as the "Half maximal inhibitory concentration" (lines 165-168).  Moreover, we've incorporated the IC50 values into the respective graphs for clarity. The calculated IC50 for each cell line is already presented in lines 228 and 229 of the original MS.

Comment 3. It is generally not recommended to use the same plots in two different figures. The plots 2A and 3A seem to (looking by eye) have same values for miR-21 and plots 2B and 3B have same values for miR34a-5p. If these are different individual experiments, this comment should be ignored. If not, the authors should setup individual experiment for the miRNA time course expression with equally spaced time points such as 0, 24 and 48 h for both cell lines.

Response 3. The similarity in values between plots 2A and 3A (miR-21) and between plots 2B and 3B (miR-34a-5p) was intentional. We utilized the same value at 24 hours to facilitate direct comparisons and evaluate the changes in expression levels between different time points, including a 0-hour time point (control, the 24-hour and the third time stamp. This allows us to better illustrate the changes in expression over the specified time intervals.

Comment 4.  The value of miR-34a-5p expressions as mentioned in line 222 does not correlate with the plot. I believe it should be 1.021 instead of 0.021 as mentioned. Similarly, the values on lines 226-227 for miR-21-5p (0.95, 0.54) do not correlate with respective plot (3A) and should be corrected.

Response 4. We apologize for this error. The fold change values are now corrected in lines 251, 255 and 256, respectively, of the revised MS.

Comment 5.  It is unclear why the authors selected for the studied miRNAs. The introduction cites a few previous studies on these miRNAs, but the relevance of these miRNAs in prostate and colorectal cancer needs can be further highlighted in consideration with the direction of study and the miRNA behavior as tumor-suppressor or oncogenic. A brief note on rationale for selecting the miRNAs in either introduction or results section can improve the report.

Response 5. In the revised MS, we have added a brief note regarding the selection of these miRNAs for the current research (lines 127-129).

References:

  1. PTEN in Colorectal Cancer: Shedding Light on Its Role as Predictor and Target Lisa Salvatore, Giampaolo Tortora et.al. Cancers 2019
  2. The functions and regulation of the PTEN tumour suppressor: new modes and prospects. Yu-Ru Lee, Pier Paolo Pandolfi et.al. Nature Reviews Molecular Cell Biology 2018

Reviewer 3 Report

I have some observations regarding the brief report titled "Thymoquinone potentially modulates the expression of key onco- and tumor suppressor miRNAs in Prostate and Colon Cancer Cell Lines: Insights from PC3 and HCT-15 Cells."

  1. "To predict the potential targets of the studied miRNAs, use the miRTarBase (https://mirtarbase.cuhk.edu.cn)" - Please cite the article for this paragraph.

  2. In Figure 1, it would be better if cell viability is expressed as the percentage of live cells rather than intensity (in). Additionally, it is necessary to perform a linear regression with the MTT data to calculate the IC50, and this should be explained in the Materials and Methods section.

  3. The section on miRNA expression analysis seems improvised for miR-34-5p, miR-21-5p, and miR-200a-5p in HCT-15, and miR-34a-5p, miR-221-5p, miR-21, and miR-17-5p in PC3. Before this, no background information is provided on why these miRNAs are important. Only miR-221 and miR-17 are mentioned. Please provide more detailed information about these miRNAs.

  4. Justification is needed for using a colon cancer cell line (HCT-15) and a prostate cancer cell line (PC3). Also, explain the relationship between colon and prostate cancer to better support the conclusions.

  5. The network that was created requires a more thorough explanation in the manuscript, including why it involves genes and biological processes. Additionally, the number of reported targets for each miRNA seems limited considering the mirtarbase repository. Please add a more detailed description of the criteria used for miRNA-mRNA network creation.

  6. It is not clear why only the expression of PTEN and BCL2 was analyzed. What criteria were used for selecting these genes? Additionally, a Western blot analysis would be appropriate.

  7. The authors should include additional experimental or bioinformatic data, specifically showing the expression of the described miRNAs in prostate and colon tumors, not just cell lines.

  8. Lines 134-135 are not clear, and the authors' idea is not well understood.

  9. Lines 151-153: At what cell confluency were the assays performed (80%)? This needs to be mentioned, and it should be clarified if the treatment times (24, 36, and 48 h) are from reaching that confluency.

A English language correction of the work would be appropriate to better understand the authors' ideas

Author Response

Comment 1.     "To predict the potential targets of the studied miRNAs, use the miRTarBase (https://mirtarbase.cuhk.edu.cn)" - Please cite the article for this paragraph.

Response 1. We are very grateful to the esteemed reviewer for his/her useful suggestions and recommendations. The citation for this paragraph has now been added to the revised MS with information (lines 191-195).

Comment 2.     In Figure 1, it would be better if cell viability is expressed as the percentage of live cells rather than intensity (in). Additionally, it is necessary to perform a linear regression with the MTT data to calculate the IC50, and this should be explained in the Materials and Methods section.

Response 2. We've now incorporated the IC50 values into the respective graphs for clarity. The calculated IC50 for each cell line is already presented in lines 228 and 229 of the original MS. The regression models of the tendencies in MTT analysis are now described in lines 169-174 of the revised MS.

Comment 3.     The section on miRNA expression analysis seems improvised for miR-34-5p, miR-21-5p, and miR-200a-5p in HCT-15, and miR-34a-5p, miR-221-5p, miR-21, and miR-17-5p in PC3. Before this, no background information is provided on why these miRNAs are important. Only miR-221 and miR-17 are mentioned. Please provide more detailed information about these miRNAs.

Response 3. The detailed background information regarding the importance of the selected miRNAs is already provided in the introduction section (lines 94-129) of the original MS. If the Hon´ble reviewer needs more information, we are ready to provide it.

Comment 4. Justification is needed for using a colon cancer cell line (HCT-15) and a prostate cancer cell line (PC3). Also, explain the relationship between colon and prostate cancer to better support the conclusions.

Response 4. These cell lines were used because they are model cell lines for evaluating multiple biological mechanisms in their respective cancers. We have now added a small justification for using these cell lines (lines 133-140) in the revised MS. We chose a colon cancer cell line (HCT-15) and a prostate cancer cell line (PC3) due to limited research on thymoquinone and miRNA interactions in these cancers (lines 362-363). Unlike other cancer types, the mechanisms in colon and prostate cancers remain less explored. Our study aims to bridge this gap and uncover new therapeutic possibilities specific to these contexts.

Comment 5. The network that was created requires a more thorough explanation in the manuscript, including why it involves genes and biological processes. Additionally, the number of reported targets for each miRNA seems limited considering the mirtarbase repository. Please add a more detailed description of the criteria used for miRNA-mRNA network creation.

Response 5. We appreciate the esteemed reviewer for his/her attention to detail. The miRNA-mRNA network creation involved a refined selection process, focusing on highly reported targets in their respective cancer types from miRTarBase. We´ve now expanded this approach, generating an updated interaction map (revised Figure 4.) encompassing a wider range of potential regulatory relationships. The criteria considered frequency, biological relevance, and experimental evidence. This expanded network addresses concerns about limited reported targets and enriches the analysis's scope and depth.

Comment 6.     It is not clear why only the expression of PTEN and BCL2 was analyzed. What criteria were used for selecting these genes? Additionally, a Western blot analysis would be appropriate.

Response 6. The selection of PTEN and BCL2 for analysis was based on their high connectivity with four of the five miRNAs studied. Due to this significant involvement, we focused on these targets to better understand their regulation, which is explained in lines 283-286 of the original MS and 286-287 of the revised MS.

Since the current journal is providing the platform to publish promising preliminary data as a ´brief report´ and since we found our preliminary data quite promising we decided to submit it as a brief report to keep our priority and hence haven´t performed any Western blot experiment. However, we will continue with this study for a deeper understanding and will keep in mind the valuable suggestion of the esteemed reviewer. We hope the Hon´ble reviewer will consider this.

Comment 7. The authors should include additional experimental or bioinformatic data, specifically showing the expression of the described miRNAs in prostate and colon tumors, not just cell lines.

Response 7. Since the current journal is providing the platform to publish promising preliminary data as a ´brief report´ and since we found our preliminary data quite promising, we decided to submit it as a brief report to keep our priority. However, we will continue with this study for a deeper understanding and will keep in mind the valuable suggestion of the esteemed reviewer. We hope the Hon´ble reviewer will consider this.

Comment 8. Lines 134-135 are not clear, and the authors' idea is not well understood.

Response 8. We have now rephrased lines 152-153 for clarity and to better convey the authors' intended idea.

Comment 9. Lines 151-153: At what cell confluency were the assays performed (80%)? This needs to be mentioned, and it should be clarified if the treatment times (24, 36, and 48 h) are from reaching that confluency.

Response 9. In the original MS, this information is already clarified (line 155) for the viability test and in the revised MS, additional information is now added in line 177.

Round 2

Reviewer 1 Report

Thank you authors for the clarifications and making the necessary modifications to improve te quality of the paper. 

Author Response

We greatly appreciate the Hon´ble reviewer´s positive feedback.

Reviewer 2 Report

Review Report 2:

1.     The authors mention that they have used 32 h timepoint for PC-3 cells in order to detect the change in miRNA expression before the doubling of cells to evaluate miRNA expression prior to doubling time (line 376-384). In that case, it is not clear why they have used 48 h timepoint for HCT-15 cells that have an average doubling time of 20-24 h. While the authors state in discussion (line 453-455) that they have followed previous studies on HCT-15 to opt for 24 and 48 h timepoints for this cell line, the different approaches used for two cell lines imply lack of consistency in experimental designs. It is suggested that the authors repeat this experiment and show the expression at all the timepoints for both the cell lines without using previous observations for Figure 3. Additionally, authors state that they intended to use the same results in plots 2A-3A and 2B-3B to facilitate direct comparisons, I believe that a repeated experiment for 24 h along with new timepoints will only improve the confidence of reproducibility of preliminary observations in this study.

2.     The timepoint of TQ treatment used for analyzing the target genes in HCT-15 cells has not been mentioned in either the description (Results section 3.4) or legend (Fig. 5).

3.     The authors have mentioned the 1.021-fold expression (also not significant) of miR-34a observed in HCT-15 cells as “slight increase” (line 371) and “mild overexpression” (line 387) in the revised MS. To my knowledge, the difference observed is very low and most miRNAs, or same miRNA in two different samples may exhibit a similar difference as observed in this study. Therefore, I suggest that any statement about miR-34a expression in TQ treatment of HCT-15 cells can only be made after repeating the experiment and observing higher change in expression or similar change with greater significance. I emphasize on this point, as this  result is most basic  to comment about any correlation between miRNA and their target gene (line 392-393). Otherwise, the discussion encompassing miR-34a-BCL2 and PTEN-miR-200a correlation can be limited to critical remarks about requirement for re-evaluation of these studies with more systematic experiments in each cell types.

4.     I suggest that it will be more appropriate to instead highlight the roles of miR-21 in colon cancer and tumor suppressive roles of miR-34a, miR-221, miR-17 and miR-21 in prostate cancer (lines 364-375, 416-453), and their other potential targets, as these are the most significant observations from TQ treatments in this study. This change in the discussion can enhance their crucial findings irrespective of the  complex results obtained from the miRNA-target gene analysis.

Author Response

Comment 1.     The authors mention that they have used 32 h timepoint for PC-3 cells in order to detect the change in miRNA expression before the doubling of cells to evaluate miRNA expression prior to doubling time (line 376-384). In that case, it is not clear why they have used 48 h timepoint for HCT-15 cells that have an average doubling time of 20-24 h. While the authors state in discussion (line 453-455) that they have followed previous studies on HCT-15 to opt for 24 and 48 h timepoints for this cell line, the different approaches used for two cell lines imply lack of consistency in experimental designs. It is suggested that the authors repeat this experiment and show the expression at all the timepoints for both the cell lines without using previous observations for Figure 3. Additionally, authors state that they intended to use the same results in plots 2A-3A and 2B-3B to facilitate direct comparisons, I believe that a repeated experiment for 24 h along with new timepoints will only improve the confidence of reproducibility of preliminary observations in this study.

Response 1. We appreciate the Hon´ble reviewer´s feedback. However, we already mentioned previously that this is an exploratory work since no reports regarding the impact of TQ on colon and prostate cancer cell lines are available so far. This article does not establish a direct correlation between these two sets of results due to the distinct cancer cell lines involved. Moreover, since the doubling times of both the cancer cell lines used in this study are quite different (HCT-15= approx. 20 h; PC-3= approx. 33 h), we believe it would not make much sense to use similar time points for both of them and hence, we explored different time points independently based on the cell types. Nevertheless, we also believe it will not invalidate the broader trends and conclusions drawn from the study.

Comment 2.  The timepoint of TQ treatment used for analyzing the target genes in HCT-15 cells has not been mentioned in either the description (Results section 3.4) or legend (Fig. 5).

Response 2. We apologize for this oversight. To provide clarity, we have now explicitly indicated that the TQ treatment occurred at the 24 h mark. This information has been incorporated in both the figure legend (Figure 5) and the Results section of the revised manuscript (line 309).

Comment 3.  The authors have mentioned the 1.021-fold expression (also not significant) of miR-34a observed in HCT-15 cells as “slight increase” (line 371) and “mild overexpression” (line 387) in the revised MS. To my knowledge, the difference observed is very low and most miRNAs, or same miRNA in two different samples may exhibit a similar difference as observed in this study. Therefore, I suggest that any statement about miR-34a expression in TQ treatment of HCT-15 cells can only be made after repeating the experiment and observing higher change in expression or similar change with greater significance. I emphasize on this point, as this result is most basic  to comment about any correlation between miRNA and their target gene (line 392-393). Otherwise, the discussion encompassing miR-34a-BCL2 and PTEN-miR-200a correlation can be limited to critical remarks about requirement for re-evaluation of these studies with more systematic experiments in each cell types.

Response 3. We appreciate the esteemed reviewer's concern about the magnitude of the observed changes in miRNA expression and their statistical significance. We acknowledge that the observed fold changes for miR-34a in HCT-15 cells were relatively low and not statistically significant. Given the limitations of further experimental validation, we have revised the corresponding statements in the manuscript to reflect the caution required when interpreting these results.  We concur that to confidently establish associations between miRNA expression and the regulation of target genes, more substantial changes in expression or higher significance are necessary. We have adjusted our discussion to remove any information suggesting a correlation between miRNAs and their targets for these specific cases in this research and focus on the need for systematic experiments to reassess these relationships in future studies, as you can find in lines 392-399 and 413-419 of the revised MS.

Comment 4.  I suggest that it will be more appropriate to instead highlight the roles of miR-21 in colon cancer and tumor suppressive roles of miR-34a, miR-221, miR-17 and miR-21 in prostate cancer (lines 364-375, 416-453), and their other potential targets, as these are the most significant observations from TQ treatments in this study. This change in the discussion can enhance their crucial findings irrespective of the  complex results obtained from the miRNA-target gene analysis.

Response 4. We appreciate the suggestion regarding shifting the focus of our discussion, which we find valuable in amplifying the impact of our findings. We have now included some potential targets that might be intriguing to investigate for those miRNAs with more notable findings in each cell line in the discussion section. Lines 399–404 for miR–34a targets, lines 432–437 for miR–221 targets, and lines 478–480 for miR–21 targets have been updated to reflect this information. Through these additions, we aim to emphasize the significance of our study's contributions while acknowledging the complexities of miRNA target gene analysis.

Reviewer 3 Report

the authors have greatly improved the article. now is suitable for publication

No comments 

Author Response

(The authors gave the same response as above.)
